# Transforming Trauma through an Arts Festival: A Psychosocial Case Study

**Jill Bennett** ⬤, **Gail Kenning** *⬤, **Lydia Gitau, Rebecca Moran** ⬤ **and Marianne Wobcke**

Big Anxiety Research Centre (BARC), University of New South Wales, Sydney, NSW 2052, Australia
* Correspondence: gail.kenning@unsw.edu.au; Tel.: +61-414332676

**Abstract:** Through a psychosocial lens, informed by relational psychoanalysis, this article discusses the design, delivery, and impact of The Big Anxiety's 2022 festival in Warwick, Queensland—an arts-based program that engages with lived experiences of trauma, distress, and suicide, and in this case with the devastating impact of youth suicide, disproportionately affecting First Nations communities. It describes the festival's methods of creative engagement, examining how these create conditions for the transformation of trauma and for experiences of growth.

**Keywords:** psychosocial; trauma; creativity; post-traumatic growth; psychoanalysis; Winnicott; design; arts festival; trauma-informed; mental health

## 1. Introduction, Rationale, and Theoretical Framework





In 2022, we developed an iteration of The Big Anxiety (TBA)—an art and mental health festival—in the regional town of Warwick in Queensland, Australia. TBA has to date been staged in three major metropolitan centres (Sydney, Melbourne, Brisbane) with smaller events in regional locations, each iteration using arts-based practices to engage directly with lived experience and the priorities of local communities. Following the Warwick festival, participants (through surveys and/or interviews[1]) reported a range of beneficial effects. Strikingly, these were frequently described as significant or life-altering shifts as in these three comments:

> *Something happened in those workshops which allowed me to focus on what I need to do to heal rather than just survive.*

> *I feel like I have gone through a journey of self-growth but mainly self-acceptance.*

> *Hope . . . was rebirthed in me. I feel like I discovered parts of me that were never allowed to flourish. So excited for the future.*

In this study, we are interested in understanding how and why such shifts were made possible—particularly after a relatively short (two-day) engagement. We describe the design of the program, its key elements, theoretical framework and approach, and how this created the conditions for change or 'growth'.

In doing so, we aim to establish how such creative, cultural programs fill a gap in mental health and trauma support. We outline the psychosocial approach of the program within which key concepts—creativity and creative apperception; holding and metabolising; and resonant form—are invoked to elaborate the particular potency of creative engagement and its relationship to growth. In line with this theoretical framework, we suggest ways of understanding life-changing effects that are not captured in conventional health metrics.

### 1.1. Filling a Gap

The Warwick TBA program initially evolved through discussions with both the local community and health care providers with the aim of filling a gap in mental health and trauma support. Held across community/cultural venues, the festival did not operate

within the rubric of healthcare or through the systems that community members perceive as "broken". Instead, it presented an opportunity to test the advantages of repositioning mental health as a cultural rather than a health/medical issue—thereby connecting with people whose needs may not be met by the healthcare system.

According to the World Health Organisation (WHO), even before COVID-19, only a small fraction of people in need had access to effective, affordable, and quality mental health care, and health systems must now prioritise "community-based mental health care [which] is more accessible and acceptable than institutional care and delivers better outcomes" (WHO 2022, p. xix). In Australia, new data confirms that over 50% of those who take their own lives have not sought formal help (Tang et al. 2022). While clearly there is a pressing need to reform clinical services, there are also compelling arguments for radically diversifying approaches to mental health. A report published in 2020 by the Australian Productivity Commission criticised the country's mental health system for being insufficiently people-focused, noting that barriers leading to poor outcomes include "a narrow view of people seeking treatment and support" and a "disproportionate focus on clinical services—overlooking [social and cultural] determinants of, and contributors to, mental health" (Productivity Commission 2020, pp. 7, 105). Other recent reports have highlighted the lack of trauma-focused support (Kezelman et al. n.d.). In Queensland specifically, the recent Parliamentary Inquiry into Mental Health (2022, p. x) recommends "a whole-of-government trauma strategy . . . that considers multidisciplinary trauma research and implements best practice strategies for responding to people that have experienced trauma".

The impacts of trauma and suicide—particularly prevalent and devastating within the First Nations community—were key factors behind the design of the Warwick program, the extent and effect of losses to suicide stressed by several participants as the reason for needing a festival:

> There are a lot of people out there who have lost their children, I mean Warwick has lost a LOT of children over the last ten years... It has been very extreme for everyone.

### 1.2. The Psychosocial Paradigm

The term 'psychosocial' describes a field of research and practice that seeks to avoid the abstraction of psychology or mental health from embodied, lived experience situated within a social nexus (Bennett et al. 2022, p. 93). By definition, a psychosocial conception of mental health takes account of social and environmental determinants and their ongoing psychological and emotional impacts. As a model of care, support, or therapy, a psychosocial approach is therefore concerned with whatever has shaped emotional/internal experience—including childhood experiences, attachment relationships, personal and intergenerational trauma histories, and circumstances such as structural and material disadvantage.

While a biomedical (or biopsychosocial) approach might potentially acknowledge social factors as inputs, it is in the field of relational psychoanalysis that the full implications of understanding and responding to experience as occurring within a 'relational matrix' are explored (Kuchuck 2021). Diverging from the traditional Freudian emphasis on individual instinctual drives, relational psychoanalysis sees interpersonal relationships as the basis of human development, and therapeutic work is therefore focused on learned patterns of interaction (Finlay 2015). In this respect, relational psychoanalysis and psychosocial studies have strong roots in Object Relations theory—and in particular, the work of the influential British psychoanalyst D. W. Winnicott.

### 1.3. Creativity

Within the Winnicottian tradition of psychoanalysis, creativity plays an emblematic role: "It is creative apperception more than anything that makes the individual feel that life is worth living." (Winnicott [1971] 2005, p. 65). This broad notion of creativity as a mode of "apperception" encompasses more than just the creation of art; it "refers to a

colouring of the whole attitude to external reality" (Winnicott [1971] 2005, p. 65). "Creative apperception" can be understood as a perceptual and sense-making process through which we know something to be true, but more importantly, experience the self as the source of truth and agency, and thereby as integrated and connected.

This conception of creativity underpins the methods and aims of the TBA workshop program. As an arts-based experiential program, the program aim is not to teach 'art making' per se but to use creative resources to create the conditions for creative apperception. The emphasis is therefore on enabling a feeling of being creative (being connected to self and truth—or to the felt sense that 'this is who I AM' (Winnicott [1971] 2005, p. 76), rather than on artistic output.

Within a relational paradigm, the definition of 'health' (the goal of therapies) is conceived in similarly experiential terms as "the ability to be, feel and act in concert with as much of who we are as possible" (Kuchuck 2021, p. 48). This emphasis on the felt connection (to self, society, and environment), which resonates with First Nations' understandings of health and wellbeing as manifested in connection to Country, people, and culture, is likewise central to the TBA program, which prioritises addressing trauma and troubling feelings of disconnection. The importance of a cultural approach to trauma is also indicated by Milroy (Zubrick et al. 2005) who notes that the concept of 'mental health' does not adequately capture the complex and multi-layered ways in which Indigenous communities experience and transmit trauma across generations, and consequently the ways in which these communities seek and find healing.

Creative apperception may be understood as a counterforce to trauma. In the aftermath of trauma (which is experienced as *done to* us and as inherently disempowering), the sense of self is profoundly affected, both cognitively and somatically (Schore 2003; Lanius et al. 2020). Further, when the self is diminished, silenced, and disempowered, and trust is eroded, "handed down knowledge is always potentially a dead end" (Phillips 2019, p. 260). The transformation of traumatic experience, therefore, requires a reconnection to the self as a self-determining or creative agent—one who can escape the oppressive requirement of "compliance" that "carries with it a sense of futility", depriving life of meaning and agency (Winnicott [1971] 2005).

The TBA program is purposefully designed to create the potential for such a creative escape or reconnection, focusing both on internal experience and on interactions with a system or 'machine' (Winnicott [1971] 2005) that reinforces the condition of powerlessness.

## 1.4. Holding and Metabolising

The cultural space of the festival workshops is a 'facilitating space' (Bennett et al. 2022) within which experience may be 'held'. As we have previously argued, facilitating environments "support the conditions to experience and metabolise discomfort" (Bennett et al. 2022). As such, they are not to be understood as static 'safe' spaces, cut off from the world, but as potential spaces in which transformation may occur as feelings are experienced and metabolised.

With this in mind, the concept of holding is paramount to our design practice and theoretical framing. For Winnicott, holding contrasts with interpretation. The latter is the verbal message that conveys information (and thereby risks being received as 'handed down knowledge'), while holding is a more subtle process, less dependent on verbal explanation (Winnicott 1960; Wright 2009, p. 31). Psychotherapists speak of 'holding space' in the sense of creating a container in which internal experience may be felt, shared, and examined without judgement. In relational psychoanalysis and associated trauma therapy, this not only creates the conditions for recognising experience but for feeling *in-relation* to another, thereby shifting the focus of therapy from the analyst's interpretation of a patient to the process of 'being with' experience and co-creating meaning or interpretation.

To the extent that it prioritises safe containment and holding as a precondition of collaborative work, the TBA program may be understood to operate according to principles of 'trauma informed care' [TIC], which foreground safety, trust, choice, collaboration, and

empowerment (Levenson 2017). TIC itself is not therapy insofar as its primary goal is to understand how experiences of distress may arise from a history of trauma and adversity, rather than to directly address past trauma (Levenson 2017).

As a cultural rather than clinical program, the TBA workshops sit somewhere between this TIC stance and a therapeutic process. The overt framing of the TBA facilitating space foregrounds creative engagement rather than therapy. The workshops do not promote an exploration of individual trauma narratives per se, nor a clinical 'intervention'. In general alignment with the model of relational psychoanalysis, they focus on collaborative process rather than content, thereby promoting what Reis (2020) calls "enactive play". In a therapeutic context, enactive play entails an engagement with felt, embodied experience and how it is felt and enacted rather than a focus on uncovering the content of trauma or what is repeated or re-lived in an enactment. As experience is enacted and metabolised, this, in turn, "creat[es] the conditions for new experience" (Reis 2020).

"New experience" in this context does not simply refer to the provision of a novel activity but to addressing the internal blocks that make having an experience—or taking in and holding onto something new—a difficult process. The goal of creative engagement in this respect is to create the potential for internal change or generate 'a new psychic structure, which allows a certain mastery over the effects of trauma' (Bollas 1992, pp. 78–79). Accordingly, participants in the TBA workshops reported accessing "buried" or unconscious aspects of experience, which lead to a shift, often perceived as "growth":

> I have gotten so much out of this. Definitely feel self-growth and a better understanding of my own emotions and mental health.

### 1.5. Resonant Form/Imagery

For Winnicott, a primary goal was to make experience feel real through recognition. The 'hole in experience' associated with developmental trauma, insecure attachment, or depression implies 'deficient containing structures' whereby experience is never matched by 'resonating forms' (Wright 2009, p. 139; Bennett 2022). The model for such matching of experience is the mother–baby relationship, in which the 'good enough' mother enacts the functions of mirroring and 'attunement' by which experience is recognised. If the therapeutic process is focused on 'holding' in a way that can address the deficiency of (maternal) containing structures and resonating forms, a process rich in sensory (audio–visual, immersive) imagery may do something similar but with a distinct practical focus, materialising resonant form to a degree that is not achieved in most verbally mediated clinical practices. Visual and material culture, in other words, offers powerful imagery or a realisation of holding to support a therapeutic or reconnective process (Bennett 2022). In the Warwick TBA program, we introduced very specific imagery of maternal holding, realised as an immersive audio–visual experience through the Road Trip workshop, described below. It is in response to this particular workshop that we see powerful 'internal' shifts described.

As a general principle, the transformational function of aesthetic form is well established; Christopher Bollas ([1987] 2018, p. 17) describes 'the uncanny pleasure of being held by a poem, a composition, a painting . . . any object', as an experience where 'self-fragmentations will be integrated through a processing form.' We are specifically interested in how people make use of resonant form to enable the integration of difficult experience, both within our image-rich, sensory, immersive program and in the everyday/creative activities that people reported following the event. In this respect, our program was designed to support the process of internal transformation—and ambitiously to test the potential for delivering this kind of outcome in a public program.

## 2. The Event: Application of Method

### 2.1. The Big Anxiety Warwick

The Big Anxiety program was initially centered around an exhibition at the Warwick Art Gallery of Edge of the Present—a mixed reality environment designed to cultivate



future thinking in place of suicidal ideation (Watfern et al. 2022; Habak et al. 2020). An accompanying two-day workshop program was designed with three integrated parts: 1. Road Trip, an audio–visual immersive experience focusing on trauma/anxiety carried in the body; 2. a creative media workshop using augmented reality (AR) and virtual reality (VR) tools and experiences; 3. a Theatre of the Oppressed workshop exploring how to bring about systemic change.

The workshop program was self-selecting and open to anyone inclined to engage within such a space. Participants ranged in age from 17 to 64. They were predominantly women, with a significant number being First Nations women, all touched by the loss of children to suicide. Some participants disclosed mental health diagnoses (including schizophrenia, BPD—Borderline Personality Disorder, and PTSD—Post Traumatic Stress Disorder). Across the diversity of experience, there was a shared sense of struggle and despair in the face of the loss of young people to suicide—and of ongoing impacts of trauma.

The program was extended at relatively short notice to encompass a Long Table on the evening before the workshops. Fifteen days prior to the launch of TBA, a 16-year-old First Nations young man died after visiting Warwick hospital seeking help for his mental health. His death—and the failure to provide effective support—had a profound impact on the community, who were left feeling "angry, sad, and more hopeless" (Courier Mail 2022). The Long Table was introduced to meet the community's need for acknowledgement and to be heard. Held in the early evening, the Long Table attracted some of those registered for the workshops but also additional people unable to attend daytime events.

### 2.2. The Long Table

*The long table event came out of a tragedy in our community recently... The Big Anxiety project was coming, so they jumped onboard with this fantastic opportunity to talk about mental health and suicide. Suicide has been happening in this community since I was a child, one of our best friends shot himself in front of his mates. He was 17 years old.*

Originally conceived by New York artist Lois Weaver, the Long Table "upends the conventional structure of a panel, which situates a few experts behind a table in front of a listening audience" (Weaver and Hunter Petree 2022). Seating 12 people, the Long Table takes the form of a dinner-party discussion in the sense that guests speak to the table, rather than addressing the surrounding audience. Initially, some of the chairs at the table are filled by people willing to seed (rather than lead) discussion. Others in the room can join the table to speak at any time. There is no pressure to speak; anyone can leave the table at any time. Discussion is not moderated. There is no intention to control the conversation, achieve specific outcomes or reach a consensus. The published etiquette states, "There can be silence. There might be awkwardness. There could always be laughter. There is an end but no conclusion" (Shaw et al. n.d.).

A neutral venue was chosen by a local community member who worked closely with the authors (JB and MW) on the set-up. A Bowen massage table was positioned by the entrance/exit so that people taking a break could make use of it (Bowen is a gentle form of massage that relaxes the body and begins to calm the nervous system (Hansen and Taylor-Piliae 2011)), and trauma counsellors/professionals were available to talk to people as they arrived or left the room at any time. Some 30 people were in attendance, along with a camera crew working on a documentary (all participants consented to the recordings, no one opted out when invited) providing opportunities for peoples' voices to be heard through recorded 'vox pops' or interviews. First Nations midwife, artist, and trauma worker Marianne Wobcke acted as host, supported by Mandandanji artist Aaron Dhuril Blades, who came from the nearby town of Toowoomba to express support.

The first two speakers to come to the table shared lived experiences as well as their general disillusionment with systems. This theme was taken up by Blades who related his own history of being caught up in an abusive justice system, after which he found 'solace' reconnecting (through his grandmother) to Country, and thereby to his people and culture. This expression of reconnection prompted a shift in energy across the room. The

affirmation that "we can lean on each other and help each other...it all starts with us" was applauded and met with further affirmations of strength; a refusal of being labelled as "broken" or "wrong", and the assertion that we are not at fault but "responding to madness of the world... feeling despair and powerlessness in face of something that is impossible to change".

Talking of her grandson's recent suicide attempt, a woman characterised the wave of suicides as a "plague" for which there is no help available—"nothing here in Warwick". Another member of the extended family then introduced herself, recounting how as a teenage mother, she had grown up with losses. Blade's story, she said, resonates with her son's. With intensifying emotion, she described the loss of her niece and how "the system let her down", and then talked more generally about the impact on children:

> *Last year was a horrible year, we lost three or four community members in their 20s. It's not stopping. We want to know what we can do to stop it!*

She referenced a very recent suicide—"the mum took him to the hospital . . . we are wanting help for them . . . we are, we are"—reinforcing as follows,

> *Because of our background[2], we are not taking our kids to be given away, because we don't want them . . . . We do all we can for them. The system's not working.*

This implicit reference to systemic racism was a catalyst for a more energised, often angry response with a series of animated comments about the need to engage families, who are being ignored, treated with suspicion, and disrespected. This elicited general agreement and a growing resolve and confidence: "we know what to do". An account of finding a pathway back through "Mob [Indigenous community] support" was applauded.

A woman came to the table and remained standing so that her words, "We're talking about doctors who are letting people down", were delivered with greater force and authority. She related her encounter with a doctor (subsequently dismissed from his role) who used racist language and interrogated her parenting skills rather than attending to the boy she had brought in.

There is a further shift in emotional tone when a woman spoke of "a very difficult journey" in her family. She recounted the suicide of her nephew who had just turned 17. In crisis, he was sent to a facility in Toowoomba where he was held for four days without seeing anyone and then sent home. This occurred a month before he took his life. The woman added "the system is *very* broken", sharing ongoing experiences of seeking help and working through 'appalling' mental health systems, which were acknowledged by the group with nods. This gave rise to a discussion of the inherent failures of 'tick and flick' approaches that permeate not only healthcare systems but education systems and governmental policies and procedures which fail to see each "person as someone important" and do not "support our kids [in] believing in themselves".

There was a growing urge to think about what *can* be done and to bring about change, affirming the power of the community:

> *I would love it,* if we have this [again] in a year's time...that [by then] everyone has a story of how they helped themselves and others . . . that is how we get the ball rolling.

As the event came to a conclusion, a woman who had been silent throughout told her story, and another walked over and hugged her; people in the room smiled, light laughter was heard as private conversations started up. The general mood at this point was uplifted and energised. Pizza was delivered, and people continued discussions in groups, often chatting excitedly. The camera crew invited anyone who wanted to record thoughts as vox pops—and a queue formed. The vox pops were notably enthusiastic and even joyful:

> *I just loved the whole thing. I've come out the other side smiling.*

> *It was great, I really liked* sharing.

*I was excited TBA was coming anyway ... I was even more excited by this. It's so exciting that you guys have jumped on this. So amazing.*

The atmosphere at the close was in marked contrast to the beginning, when nervous tension was evident, and a reluctance to take part was aired, "We are here just to look we don't want to take part ... ". Reflecting on the evening, some acknowledged they had pushed through feelings of discomfort to engage; a few people left the room overwhelmed by emotion but were then motivated to return. People lingered in conversation for some time, and it was almost two hours after the close of the Long Table when the last people left.

### 2.3. Two-Day Intensive Workshops

The Long Table led into two days of participatory workshops in three interlocking parts:

### 2.3.1. Road Trip

Drawing on her background as an Indigenous trauma-focused midwife, nurse, and artist, Marianne Wobcke led participants in her 'Perinatal Dreaming' workshop or 'Road Trip' on the morning following the Long Table. Perinatal Dreaming explores pre-peri and post-natal experiences with an emphasis on cultural connection and repair of trauma and experiences of disconnection. The framing of the Road Trip blends perinatal imagery with an implicit understanding of developmental processes of attachment and maternal holding. Marianne herself hails from the lineage of 'Stolen Generations' of Aboriginal children forcefully removed from their families at birth under the racist policies of successive Australian governments (Wobcke and Bennett 2022). In this regard, 'Perinatal Dreaming' strongly resonates with the experiences of First Nations people described in the Long Table. It aims to break the cycles of trauma originating in colonial violence and to re-establish connection with 'Country'[3] through a 'Dreaming' story[4].

In her preamble, which is both playful and clear about how the process can be effective, Wobcke acknowledges people's experiences of pain and trauma, addressing the potential for 'triggering' as participants became conscious of trauma held and emerging from within and guiding participants to be self-reflective and aware.

The Road Trip itself is an immersive audio–visual experience, structured through a carefully curated playlist of music to take participants through the stages of birth/creation, experienced as an embodied journey. The sense of a 'trip' promotes a feeling of moving through different phases (some difficult) with the potential for release or catharsis. Accompanying imagery is screened through large-scale projection, while participants lie on mats on the floor or move around as they wish. Each track coincides with a stage of birth: 'The Call to Adventure'; 'Conception', connecting participants to Country; the 'Good Womb', the 'Toxic Womb'; then 'Labour begins' and 'intensifies', and a period of 'Transition' characterised by 'Chaos' follows. This is followed by a 'Death/Rebirth Experience', first as 'Triumphant' then as 'Struggle'. The music then leads participants to 'Home again: Connection to the Sacred', then a sense of 'Oceanic Bliss', and 'Re-inspiration'. As the session draws to an end, there is period of silence, in which participants sit in stillness, experiencing 'Dadirri' (deep listening). The Road Trip ends with nature sounds, 'Kanyini', reconnecting participants to Country.

The aesthetics of the Road Trip and its bridging of Indigenous spirituality with psychoanalytic understandings of trauma and attachment theory will be discussed in detail elsewhere. Here, it is important to note that it is the aesthetic (sensory, affective, embodied) experience rather than specific content that is operative in terms of generating a sense of 'holding' and/or transitioning through phases of struggle.

Noticeably, it is the structure and flow of the Road Trip that impacts people—regardless of their preference or otherwise for the tracks played:

*I remember... standing at the window with my back to everybody, shutting out everyone there, staring out the window, crying at the music because every single piece of music elicited a subconscious response. It was almost a primal response.*

> *. . . obviously, that was huge. Marianne's journey . . . it was intense. I remember sitting, lying there actually and thinking, music, it's not going to affect me. I'm stone cold, tough, and just being like, that was a whole journey that was full on*

### 2.3.2. Creative Media

Following the Road Trip and debrief, an informal Creative Media workshop introduced tools for creative engagement with emotions—such as EmbodiMap, a VR tool informed by somatic trauma therapies that enables users to engage with and map their feelings, thoughts, and emotions and how these are experienced within the body (Gitau et al. 2022; Burgess 2022).

Other works available included Waumananyi, a VR experience co-created by Ngangkari (healers) and Anangu (First Nations people from the Central Desert region) artists, and Uti Kulintjaku, which evokes experiences of being mentally trapped and disempowered—and then released and held—through the traditional story of a man who becomes trapped in a log. As discussed elsewhere (Bennett 2022), the log serves as a 'holding space' for feelings of distress, as well as generating a sense of hope for transformation at the end when the man is set free and reconnected to community. Participants also engaged with Hard Place/Good Place, an AR project focused on telling stories of being in a hard place/good place, for which they then had the opportunity to generate stories. This element was part of a strategy for continuing creative engagement with the community.

### 2.3.3. Forum Theatre

On day two, the focus on embodied trauma shifted to a collective focus on external environments and creating change. Minola Theatre and Arts for Inclusion facilitated a Forum Theatre workshop built on the principles of the Theatre of the Oppressed [TO], developed by the Brazilian theatre maker Augusto Boal. TO uses theatre as a means of promoting social and political change, enabling actors from 'oppressed' communities to explore the ways in which oppression is played out. TO aims to empower actors and inspire individual and collective transformation (Boal 1979; Singhal 2004), in this case, by uncovering and disrupting the power relations inherent in interactions within mental health systems.

The workshop begins by using playful activities to explore storytelling, explicitly inviting people to stretch themselves within a 'doughnut of discomfort' (or 'window of tolerance' (Siegel 1999)), referring to the range of intensity of emotional experience that can be comfortably experienced, processed, or integrated.

Using the analogy of a dysfunctional machine, participants were asked to think about—and then collaboratively re-enact—what is not working in the mental health system. Participants drew on many themes that surfaced in the Long Table (box-ticking, lack of connection, silencing, reliance on pills, diagnosis, referrals) to produce a physical re-enactment that served to contain their frustrations with a system that one participant summed up with the phrase "Shut up and let me help you!" This was followed by an exercise enacting 'what works', through which participants identified and performed being heard, government support, harmony, and giving time—thereby jointly manifesting a Machine for Change. The latter facilitated the expression of hope and practical ideas for how things can be improved.

This exercise was particularly resonant for one participant who had been vocal in the Long Table about the system being broken:

> *I think the thing that still sticks with us so powerfully was the circle work we did when we did the Machines . . . that is still probably the highlight. It was just so moving! . . . knowing that something as simple as that could have such a powerful effect on people, it's stuck with me.*

## 3. Outcomes

### 3.1. Design and Facilitation

In retrospect, the Long Table played a critical role in establishing trust and focusing the intent of the community:

"The long table was a great beginning, it helped me feel safe and heard. For me it set the stage for the next two days. By the end of the program, I was floored by the safe space that was created and the hope that was rebirthed in me. I feel like I discovered parts of me that were never allowed to flourish. So excited for the future. Thank you."

If the Long Table initiated the process of holding so that people felt "valued, safe and heard", this built up over the following two days with the effect that some participants articulated a gradual increase in receptivity ("Today I was more open and receiving of information").

In its open format, the Long Table embodies light touch and responsive design that attends to the whole environment as both a psychosocial and material space. Its facilitation entails attentive listening and attunement to the emotional valence and flow of the discussion. In this sense, we see the process as one of specialised (trauma-informed) psychosocial design, a large part of which relies on pre-engagement with—and attunement to—the community. As Marianne said in the debrief:

"We brought ourselves to the table with the message, we're here, willing to sit in uncertainly. We're ready to listen. This is your community, your knowledge. We can handle your rage, your fear, whatever you bring. We are not scared to step into this with you. You are not too much!"

This approach was maintained throughout the program with an emphasis on low-key facilitation. While workshops are led by facilitators, the participants commented on the relative lack of hierarchy and the fact that there was often no distinction between staff and participants. Similarly, at the outset, a trauma counsellor was identified as being on site and available for debrief/support. One participant joked at the end of the event that she made a note to avoid the counsellor—a comment that in context reflected both the atmosphere of trust and connection, and the growing capacity of the group to sustain itself through mutual support.

### 3.2. The Experience: A Participant Story

*Probably the best thing I've done happened on the night of the first workshop. I was able to look at [my daughter's] photo on our wall. Look into her eyes, connect with her. I felt comfortable looking at her photo for the first time since she died.*

To demonstrate the impact of creative engagement, we focus on the case of participant A, who came to the workshops with a degree of scepticism but then described and documented a process of change in the wake of debilitating grief following the loss of her daughter:

*Since my daughter killed herself 30 months ago, I've been frozen in grief, pain. Suddenly, things had no meaning. I had to rethink everything. She's gone and I'm broken.*

Part of this entailed a withdrawal from social interaction:

*Mentally, I was in a holding pattern of 'if I just stay away from everybody and if I just sort of sit here in the office, I'll be fine'. I had made it very clear I wanted no interaction with people.*

Invited to The Big Anxiety workshops by a friend, participant A's initial thoughts were, "I have no idea what it's about, but I'll go check it out". She was at first sceptical of the Road Trip:

*I remember during the day thinking, oh my God, this is just hippie shit. But I went, you know what? I'm going to go with it. Just give over.*

On 'giving over', she experienced an emotional response, finding resonance with the Road Trip music:

> *I remember when it was... I was standing at the window with my back to everybody, trying to shut everyone out, staring out the window, crying at the music because every single piece of music elicited a subconscious response. It was almost a primal response. And I could feel it like just right there (sighs deeply).*

The playlist of the Road Trip is in fact an eclectic mix of popular songs that, in themselves, would be unlikely to appeal to everyone's taste. In this context, however, it is the role of each track and what it represents or does (in terms of holding, releasing, moving forward, and so on) within the whole experience that is significant. As such, the function of the Road Trip is not to promote the appreciation or enjoyment of music per se but an experience of creative apperception—of making use of an audio–visual experience to feel a connection to the self. In participant A's case, this gave rise to both a direct expression of creativity and an experience of connection to the group.

> *I came home and just picked up a book and I just wrote a poem! A crappy poem, but I wrote it and it had meaning for me. And then the next morning I bounced up out of bed, and I, oh, I went spring! I went there, and I felt, I felt awesome. They had the butcher's paper up, and I went round and wrote on every single one of them, said good morning to every single person (laughing). Then I came across and read my poem for people at the end of the day.*

Participant A describes an awakening: "almost like a gentle slap to say, hey wake up! And it was okay, it was fine". In her reflection after the festival, she speaks of the importance of "the ability to have choice and control" and "permission to grieve how you want" after a suicide:

> *That's what I was looking for. I was looking for someone to say, it's okay.*

In this sense, the event functioned for her as a facilitating space to realise this potential. The poem, titled "The Storm," written six weeks after the festival and later read out publicly by her, provides insight into how she found this 'permission', and the ways in which she felt held and came to process deep negative emotions. Part of this poem reads:

> *I walked into the storm though I knew I may not make it back.*
> *I had to feel the sting of the rain on my skin: tattoo needles of pain.*
> *Hear the wind howl as I added my screams ripped from within daring mother nature.*
> *To match my anger . . . .*
> *Take it all I screamed, ALL.*
> *... I am a child with no parent to hold me close . . .*
> *The storm was around me and through me, wrapped me up in tumultuous arms and*
> *Became me...*
> *The fight was done.*
> *The storm was over.*
> *I walked out, the thunder abated in my soul.*

## 4. Reflections

### 4.1. Post-Traumatic Growth and Connection to Self

Participant A's story embodies the reconnection to self that grounds "the ability to be, feel and act in concert with as much of who we are as possible" (Kuchuck 2021, p. 48) that is the marker of health. Her felt connection clarifies a pathway to healing:

"Something happened in those workshops which allowed me to focus on what I need to do to heal rather than just survive. I'm thinking lots and I know the undoing of our history; our healing will take a long time. But we have to do it."

This shift may also be understood in terms of Post-traumatic Growth (PTG), defined as the ability to learn from and thrive after traumatic experience (Tedeschi and Calhoun 2004; Zoellner and Maercker 2006). PTG is demonstrated by an increased appreciation for life, more meaningful interpersonal relationships, an increased sense of personal strength, changed priorities, and a richer existential and spiritual life (Tedeschi and Calhoun 2004).

This article is limited to examining the experience of the initial Warwick workshops and their immediate effects (the longer-term impacts will be discussed elsewhere). Hence, we cannot comment here on sustained change but can point to an orientation to PTG, evinced in the resolve of multiple participants, their sense of being enlivened (for example: "Uplifted, reconnected with myself and life") and in reports of changing relationships—such as Participant A's account of connecting with the group, and then her reengagement with the photographs of her daughter. There is also this lucid account by another participant who describes the immediate impact on family relationships:

> When I walked away from going to the Big Anxiety in Warwick, it was amazing! Like I was on absolute high and just felt like for the first time in a long time, that there was a place I was supposed to be, you know, and that I was heard ... from that moment I was like, I've come home! I needed that so much. And just to be around such supportive people and people that want to make a difference. I think that was a huge shift. ... I walked away from that feeling like I had a voice and that my voice was important. And whether people want to listen to it or not, it's important that I speak my truth. I'll be true to who I am.

> It was such a great opportunity to be part of that because for a two-day workshop, to be able to change my outlook on life... I can't even explain the immense change that's made. And I think not only for me, but for my little family as well, like to be able to sit down and even address issues with my own daughter and her anxiety at eight years old, being able to sit there and use the skills learned from being at the Big Anxiety to be able to open that conversation with her ... it's definitely made a huge change. That's for sure.

More than once, the experience is described as one of rapid growth that feels therapeutic in effect. Tellingly, here, it is conceived as a process of 'work' rather than simply leisure:

> Working with The Big Anxiety, I feel like there's been maybe ten years of growth in a short span of time ... that you maybe wouldn't even get in a lifetime.

The offering is also understood as one of healing that fills a gap in service provision:

> This heals. It's tailored to everyone because it's you doing it for yourself. The gap this fills is a very big gap in the market.

We suggest that part of the reason that people felt that 'work' had been done was the depth of engagement with challenging, as well as good feelings—itself a function of a holding process that enables safe exploration. This was explicit in the Long Table, but then also in the Road Trip, which facilitated a journey through negative and positive experiences (with imagery evoking the good/toxic womb, good/toxic breast and so on, with a return to Country and connection). In this sense, the workshops enabled what theorists of PTG refer to as the coactivation of positive and negative emotions, which provides fertile ground for optimal growth and development (Hartman and Zimberoff 2007, p. 4). Notably, "effective coping with major traumas requires dealing with and working through much more negative information and, hence, is associated with a higher proportion of negative thoughts and emotions" (Larsen et al. 2003, p. 216; Hartman and Zimberoff 2007). Good facilitation in this context enables "tolerating the negativity long enough to process it and to benefit from the processing" but also enabling simultaneous positive experiencing. This was not only facilitated through verbal interaction and holding but was effectively orchestrated and visualised in the Road Trip and in other experiences (including VR artworks), which activate transition in a way that can feel awe-inspiring and exhilarating without allowing a manic positivity to obscure the need to sit with and process negative feelings. Thus, participants often reported dealing with or processing difficult material but landing well:

"Com[ing] out the other side smiling even though it is such a difficult topic is really exciting and really powerful."

## 4.2. Community

Finally, however, it is important to stress that the TBA program is not repeated in a fixed format but co-produced and tailored to meet the needs of specific communities. The positive impacts achieved in Warwick arise from a particular confluence of community intent and workshop practice, grounded in a shared acknowledgement of the need for action (in relation to suicides, as per above). The participants in this case were notably motivated for change. The community member working with us on recruitment observed that "we got the right people here". Accordingly, a number of participants echoed the sense that there were important outcomes at a community level:

*I loved it. I know that I can go to bed tonight knowing that there is some traction about how this community can move forward...*

*Really inspirational to see people from community come together . . . knowing that we have installed a little bit of courage and strength in this community to have a say in what happens—towards meaningful change rather than the tokenism that the government put up.*

*I feel like everyone really discovered strength in the community and that moving forward they will all act on and use for the best for the community.*

*I found that we are not alone . . . we are going to be able to work together to build stuff... and to keep going with community members to find solutions.*

In practical terms, there was a sense that TBA was an enabling process, offering tools for change:

*. . . it's great to have a show come to town, but if we can actually pick those tools up and put them into practice, and actually utilize them... for me, the fact that they [TBA] want to come and they want to teach us and they want us to take this from a grassroots level and make it part of our community, but not them doing it, us doing it because we know the community, that's my big drawcard. I'm learning new skills.*

This sense of "us doing it" is perhaps the most important indicator of sustained change, on which we will report elsewhere. For the research team, it underlines the importance of a model of facilitation that prioritises listening to and responding to lived experience, not only during but pre- and post-engagement. Such responses highlight the degree to which a sense of community empowerment, combined with capacity-building, may prove to be the key to extending and maintaining positive outcomes. Immediately following the festival, we heard that participants had formed a social group and were meeting regularly with the aim of building on the festival program. They were also motivated to travel to The Big Anxiety's festivals and courses in other cities over the following year. In this light, future research may consider the question of whether and how the community itself takes on the function of a holding structure.

This community-led orientation and focus on collaborative creativity does not imply a lesser focus on creative technique, training, and skill. On the contrary, we have demonstrated the need for an expanded conception of psychosocial design that is both skilful and adaptive, and for the need to provide creative resources, programs, or 'tools' that can be accessed and utilized by communities. Further analysis will also look at the continued use of the VR/media tools provided. The psychosocial case study we have offered may be understood as an evidence base informing the use and iterative development of such resources—and as a model for developing the resource base that we need for a genuinely trauma-informed approach to wellbeing.

**Author Contributions:** Conceptualization, J.B., G.K., L.G., R.M. and M.W.; methodology J.B., G.K., L.G., R.M. and M.W.; formal analysis, J.B., G.K., L.G., R.M. and M.W.; investigation, J.B., G.K., L.G., R.M. and M.W.; writing—original draft preparation, J.B., G.K., L.G.; writing—review and editing, J.B., G.K., L.G., R.M. and M.W. All authors have read and agreed to the published version of the manuscript.

**Funding:** This research was funded by the Australian Research Council through a Laureate Fellowship awarded to Professor Jill Bennett, FL170100131 645.

**Institutional Review Board Statement:** This HREC is constituted and operates in accordance with the National Health and Medical Research Council's (NHMRC) National Statement on Ethical Conduct in Human Research (2007). The processes used by this HREC to review multi-centre research proposals have been certified by the National Health and Medical Research Council. This study was approved under HC17649.

**Informed Consent Statement:** Informed consent was obtained from all subjects involved in the study.

**Data Availability Statement:** The data presented in this study are available on request from the corresponding author. The data are not publicly available due to the potential risk to participants becoming identifiable.

**Acknowledgments:** We acknowledge and thank the community and participants of Warwick, Queensland.

**Conflicts of Interest:** The authors declare no conflict of interest.

## Notes

[1] The event survey invited open-ended narrative responses. Interviews were semi-structured but open-ended, conducted primarily on video at the end of the festival. Mixed qualitative methods were employed, with a focus on phenomenological analysis.

[2] The 'background' referenced here is in relation to being First Nations people whose children were historically taken from them by successive Governments enacting a racist policy of assimilation see https://aiatsis.gov.au/explore/stolen-generations.

[3] Connection to 'Country' is key to First Nations people's existence and identity. 'Country' is a living entity that gives and receives life. It is home, peace and nourishment for body, mind and spirit D.B. Rose (1996).

[4] 'Dreaming' is the word used to explain how life came to be in Aboriginal culture (Stanner 1979).

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
