# Peer review of "Transforming Trauma through an Arts Festival: A Psychosocial Case Study"

_socsci, doi:10.3390/socsci12040249_

Round 1

Reviewer 1 Report

This is a strong and engagingly presented piece of research that accounts for the authors' experience of designing, leading and evaluating The Big Anxiety Festival in Warwick, Queensland. The scope of this article is ambitious (and potentially unwieldy), but the authors deftly present the background, rationale and choices to ensure that its guiding theories, methods and goals are quite clear and persuasive. Winnicott's ideas of creative play and holding are initially offered as key referents for understanding the work of the festival, although these recede in focus as the study progresses to focus on specific workshops/formats. 

The authors seem aware of the risks of making too-strong claims for this work -  its positive effects could be short-lived if not part of a sustained programme of work or treatment for those involved. However, the article could still be enhanced by a few additions that might speak to these concerns or potential questions by readers:

1) What is the status of the festival with local mental health service providers   - are they aware of it, do they follow up with the participants or organisers in any way? How do the festival organisers try to fill the 'gap' identified beyond the festival - is there any kind of debriefing with mental health services? Or are there structures to sustain the positive effects? How does the festival function as part of broader advocacy?

2) A good deal is claimed about creativity and holding as an effect of the festival, but I wondered too about the issue of community; and even community as a holding structure. How was the creation of community, however temporary, important for the experience of success? 

3) How were/can those targeted by the festival be invited to help advise on future iterations of the format? The article claims that the 'positive impacts achieved in Warwick arise from a particular confluence of community intent and workshop practice, grounded in a shared acknowledgement of the need for action' (p.13), but it's not clear who is making all these decisions and determinations, or their precise relationship to First Nations people. 

Not all of these points require an intervention in the current text, but I offer them as provocations that might strengthen its claims. 

I wish the authors well with the revisions, and look forward to seeing the article published. 

Author Response

We thank reviewer one for their engagement and support of this paper. The article could still be enhanced by a few additions that might speak to these concerns or potential questions by readers:

1) What is the status of the festival with local mental health service providers   - are they aware of it, do they follow up with the participants or organisers in any way? How do the festival organisers try to fill the 'gap' identified beyond the festival - is there any kind of debriefing with mental health services? Or are there structures to sustain the positive effects? How does the festival function as part of broader advocacy?

Mental health service providers were attendant at the festival, but not in a formal capacity, and not to an extent that warrants focus in paper  - We have addedd a paragraph to talk about what happens afterwards and a second paper is in production that addreesed this in more detail

 Such responses highlight the degree to which a sense of community empowerment, combined with capacity building may prove to be the key to extending and maintaining positive outcomes. Immediately following the festival, we heard that participants had formed a social group and were meeting regularly with the aim of building on the festival program. They were also motivated to travel to The Big Anxiety’s festivals and courses in other cities over the following year. In this light, future research may consider the question of whether and how the community itself takes on the function of a holding structure.

2) A good deal is claimed about creativity and holding as an effect of the festival, but I wondered too about the issue of community; and even community as a holding structure. How was the creation of community, however temporary, important for the experience of success? 

We feel that these important points about community are also addressed by the paragraph above

3) How were/can those targeted by the festival be invited to help advise on future iterations of the format? The article claims that the 'positive impacts achieved in Warwick arise from a particular confluence of community intent and workshop practice, grounded in a shared acknowledgement of the need for action' (p.13), but it's not clear who is making all these decisions and determinations, or their precise relationship to First Nations people. 

The last author on this paper is First Nations trauma background and worked on the festival in this capacity as indigenous led researcher

Reviewer 2 Report

This is a well-writen and engaging paper that fits in very well with the brief of the special issue call. The authors focus on the psychosocial approach and provide a good overview and description of approaches used and the theoretical underpinning.

The authors mention that his is a case study and provide a number of examples. I would like to see some more "traditional information" so that the reader can better make sense of what is presented: this includes how many people participated, what type of data was collected and how, and how the data was analysed (or more specifically, how the quotes presented in the paper were selected). Also a reflection on who was not there from this particular community and what this might mean for the approach. 

The authors point out that the article focuses on the impact of the intervention and its immediate effects and that long-term changes are (or will be) reported elsewhere (p.12). However, a few sentences, critically  reflecting on how this approach can/cannot impact on the wider social context and resources would be welcome.

Finally, a short reflective section on the authors and their roles/backgrounds would be very welcome. For example, are the authors first nation representatives? what is their link to the community they work with and their professional backgrounds?

Author Response

We thank reviewer two fro their support of this work.

We have added information regarding the data collection methods and analysis approach.

The event survey invited open-ended narrative responses. Interviews were semi-structured but open-ended, conducted primarily on video at the end of the festival. Mixed qualitative methods were employed, with a focus on phenomenological analysis.

Also a reflection on who was not there from this particular community and what this might mean for the approach - we acknowledge the importance of reflection on who was not present and feel this can be better engaged with in an article in production regarding follow on activities

The authors point out that the article focuses on the impact of the intervention and its immediate effects and that long-term changes are (or will be) reported elsewhere (p.12). However, a few sentences, critically  reflecting on how this approach can/cannot impact on the wider social context and resources would be welcome. We have added

Such responses highlight the degree to which a sense of community empowerment, combined with capacity building may prove to be the key to extending and maintaining positive outcomes. Immediately following the festival, we heard that participants had formed a social group and were meeting regularly with the aim of building on the festival program. They were also motivated to travel to The Big Anxiety’s festivals and courses in other cities over the following year. In this light, future research may consider the question of whether and how the community itself takes on the function of a holding structure.

Finally, a short reflective section on the authors and their roles/backgrounds would be very welcome. For example, are the authors first nation representatives? what is their link to the community they work with and their professional backgrounds?

While we have not provided individual profiles of authors for this article, it is apparent now in the naming of the authors and their role in the festival, that it is an indigenous-led project with first nations author taking that lead in the festival.

Many Thanks

Reviewer 3 Report

This manuscript describes the impact of an event in which artistic manifestations were listed in its programming with the development of strong and vivid memories of trauma. It describes the methods of creative engagement of the festival, examining how they create conditions for the transformation of trauma and growth experiences. In the way that Winnicottian psychoanalysis brings creativity as essential to human existence at the same time as “having life as something worthwhile”, art is also seen here as something that takes place in a festival that aims not to fit Someone to elaborate any work that is not in itself the transformation of oneself and the perception of existence in some sense in living.

The topic is interesting and it was properly approached by the author I.

Author Response

We thank reviewer three for their support and not their comments with interest. 

This manuscript describes the impact of an event in which artistic manifestations were listed in its programming with the development of strong and vivid memories of trauma. It describes the methods of creative engagement of the festival, examining how they create conditions for the transformation of trauma and growth experiences. In the way that Winnicottian psychoanalysis brings creativity as essential to human existence at the same time as “having life as something worthwhile”, art is also seen here as something that takes place in a festival that aims not to fit Someone to elaborate any work that is not in itself the transformation of oneself and the perception of existence in some sense in living.

The topic is interesting and it was properly approached by the author